# Bulb Dormancy In Vitro—*Fritillaria meleagris*: Initiation, Release and Physiological Parameters

**DOI:** 10.3390/plants10050902

**Published:** 2021-04-30

**Authors:** Marija Marković, Milana Trifunović Momčilov, Branka Uzelac, Slađana Jevremović, Angelina Subotić

**Affiliations:** Department of Plant Physiology, Institute for Biological Research “Siniša Stanković“—National Institute of the Republic of Serbia, University of Belgrade, Bulevar Despota Stefana 142, 11060 Belgrade, Serbia; milanag@ibiss.bg.ac.rs (M.T.M.); branka@ibiss.bg.ac.rs (B.U.); sladja@ibiss.bg.ac.rs (S.J.); heroina@ibiss.bg.ac.rs (A.S.)

**Keywords:** dormant period, fritillary, Liliaceae, sprouting

## Abstract

In ornamental geophytes, conventional vegetative propagation is not economically feasible due to very slow development and ineffective methods. It can take several years until a new plant is formed and commercial profitability is achieved. Therefore, micropropagation techniques have been developed to increase the multiplication rate and thus shorten the multiplication and regeneration period. The majority of these techniques rely on the formation of new bulbs and their sprouting. Dormancy is one of the main limiting factors to speed up multiplication in vitro. Bulbous species have a period of bulb dormancy which enables them to survive unfavorable natural conditions. Bulbs grown in vitro also exhibit dormancy, which has to be overcome in order to allow sprouting of bulbs in the next vegetation period. During the period of dormancy, numerous physiological processes occur, many of which have not been elucidated yet. Understanding the process of dormancy will allow us to speed up and improve breeding of geophytes and thereby achieve economic profitability, which is very important for horticulture. This review focuses on recent findings in the area of bulb dormancy initiation and release in fritillaries, with particular emphasis on the effect of plant growth regulators and low-temperature pretreatment on dormancy release in relation to induction of antioxidative enzymes’ activity in vitro.

## 1. Introduction

Geophytic species are plants with a life-form in which the perennating bud resides in an underground storage organ [1]. Geophytes are usually referred to as species with a very short aboveground growth period, whereas in the unfavorable period of the year, they survive in the form of specialized underground storage organs—bulbs, corms or tubers. After the period of active growth and flowering during spring, the senescence of the aboveground tissues followed by root senescence occurs, while the plant enters a dormant period without visible organogenesis. Dormancy is an adaptive trait that allows plants to maximize success by increasing the possibility that seed germination and/or vegetative growth occurs in the most advantageous season [2].

*Fritillaria meleagris* L. (Liliaceae), also called snake’s head fritillary, is a valuable bulbous ornamental plant. The genus *Fritillaria* includes about a hundred species, mainly found throughout temperate parts of the Northern hemisphere. *F. meleagris* has attractive purple-white flowers, rendering this plant very desirable in horticulture. *Fritillaria* species are very often cultivated as ornamental garden plants and their bulbs have been used in traditional medicines of China [3], Turkey [4] and Japan [5]. A number of *Fritillaria* species contain specific alkaloids that are of interest for the pharmaceutical industry because of their antitussive and expectorant properties. Production of fritillaries by conventional methods (by seeds) is very slow and takes up to several years to produce the whole plant [6]. Therefore, from economic and horticultural perspectives, studies related to propagation of this plant are very important, especially regarding dormancy: its initiation, breaking and physiology.

Since many geophytes are considered as ornamental plants, there is an increasing commercial interest in their propagation by vegetative reproduction. Therefore, dormancy release and early sprouting are important goals in horticulture. This particularly applies to *F. meleagris*, a very attractive popular ornamental species. The majority of ornamental geophytes have a very low natural propagation rate. Their multiplication is based on splitting of mother bulb and production of daughter bulbs. Only a few new bulbs can be regenerated every year from the mother bulb. Apart from slow propagation, another persistent problem related to reproduction is the source of plant material for further propagation. Namely, using only one bulb as a starting material for obtaining new bulblets increases the possibility of viral and fungal infection, which could be transferred to daughter bulbs. Therefore, numerous propagation techniques have been developed to obtain healthy material and to speed up breeding and sprouting of new bulbs [7]. On the other hand, in vitro techniques have been used to clarify physiological aspects of dormancy and processes associated to it. Micropropagation methods of *Fritillaria* species, with special attention to *F. meleagris* and approaches for overcoming bulb dormancy, are reviewed in detail by Petrić et al. [7].

## 2. Dormancy

Dormancy occurs during the life cycle of geophytes and is necessary for their normal development [8,9]. Bulbous species develop dormancy to survive unfavorable environmental conditions, with different species responding differently to diverse environmental factors. Together, environmental factors, as well as species-specific responses to these factors, determine when bulbous plants enter or exit dormancy. These environmental factors include temperature, photoperiod and water availability [9]. 

In dormant seeds and vegetative organs, no external morphological changes can be detected. However, various physiological and morphological changes occur inside dormant bulbs, such as differentiation of floral buds or roots [10,11]. 

Specific factors influence the duration and termination of dormancy in bulbs: specific proteins [12,13], changes in the level of gibberellins and abscisic acid [9,14,15,16,17,18,19], amylase-dependent degradation of starch [20,21,22,23], as well as hydration of large molecules and subsequent water release from hydrated molecules [23,24,25,26]. In certain cases, dormancy release leads to an increase in cell division rate [27]. 

The mechanisms inducing dormancy are not fully elucidated. Environmental factors such as air and soil temperature, decreasing soil moisture and photoperiod play key roles in the initiation of dormancy [1]. During regeneration in vitro, bulbs and other storage organs (tubers and corms) usually develop dormancy, as in natural conditions. Growth and sprouting of geophytes in vitro are influenced by dormancy, bulb size and maturity [28]. Consequently, an efficient in vitro protocol, including dormancy release, is desirable for fast, efficient and valid commercial exploitation of all horticultural geophytes. Most studies investigated micropropagation by bulb formation in tissue culture. In fritillaries, these studies were focused on optimization of nutrition media and culture conditions [29]. Reproducible in vitro regeneration of many *Fritillaria* species required plant growth regulators (BAP, indole-3-acetic acid (IAA), NAA, dichlorophenoxyacetic acid (2,4-D)) at concentrations ranging between 0.1 and 1 mg L^−1^ and exposure to low temperatures (4–15 °C) [7]. 

The entire physiology of dormancy, from its induction to termination, is under hormonal, molecular and environmental control. At present, the molecular mechanisms of regulation of dormancy initiation and release still remain unclear and least elucidated. Early attempts to isolate genes expressed during bulb formation aimed at clarifying the relationship between the induction of bulb formation and the induction of bulb dormancy [30,31]. Transcriptome sequencing studies concerning screening of differentially expressed genes associated with low temperature laid the foundation for exploring the molecular mechanism of low-temperature-induced dormancy release [32,33].

### 2.1. The Role of Temperature in Bulb Dormancy Initiation and Release

Of all environmental factors, temperature is considered to play a crucial role in controlling growth and flowering in geophytes, most of which require a “warm–cold–warm” sequence to complete their life cycle [34]. Bulbous plants (in temperate regions) require an extended period of low temperature during the year. The absence of this cooling period, of particular duration, leads to very slow shoot growth and severe flowering disorders [35]. High temperatures induce dormancy, while low temperatures break it. Dormancy in most species of the genus *Fritillaria* is initiated in the summer. During the winter months, low temperature enables dormancy release and sprouting in the early spring of the following year [7,29]. In controlled in vitro conditions, the application of a particular temperature regime can replace natural seasons. Such conditions allow us to study the influence of temperature on dormancy initiation and release by monitoring changes in bulb characteristics during the time course of a particular temperature regime, which has to be determined for each plant species. Fritillary bulbs regenerated in vitro could not continue to grow and develop normally without having been subjected to low temperature, similar to many other geophytes, such as corydalis [36]. 

Numerous in vitro studies showed that *Fritillaria* bulbs and seeds must be subjected to low temperature for dormancy breaking and subsequent sprouting and flowering. In *F. meleagris* seeds, low temperature (10 °C) for 10 weeks was required for dormancy release [7]. The effects of temperature regime and nutrition media composition on bulb dormancy release in vitro in some *Fritillaria* species are summarized in Table 1.

Bulbs of *F. meleagris* regenerated in vitro became dormant and ceased growing, sprouting and regenerating leaves, if they had not been exposed to low temperature [39]. Bulbs, therefore, were not able to develop, grow and sprout properly even under favorable conditions for the overall plant growth. As a result of dormancy, bulbs of *Fritillaria pallidiflora* Schrenk stopped growing and forming leaves [43]. Low temperature could break dormancy in vitro and increase the percentage of sprouting and regeneration of new bulbs [40]. Treatment with low temperature was crucial for almost all fritillaries, but the effective temperature and its optimal duration differed among species (2–15 °C for 4–12 weeks was effective in most fritillaries) (Table 1). Multiplication rate could be increased when initial explants (bulb scales or parts of bulbs) were exposed to low temperature for several weeks before induction of in vitro morphogenesis.

Temperature affected the number of bulbs as well as their weight. When bulbs of *F. meleagris* were exposed to low temperature (4 °C) for 8 or 10 weeks, they did not multiply, unlike when they were cultured at 15 and 24 °C [39]. The cold pretreatment lasting for 6 weeks resulted in a weight increase of new bulbs regenerated in culture (Figure 1). Low and moderate temperature (4 and 15 °C), extended for over 6 weeks, led to a bulb weight increase compared to control cultures grown at 24 °C. The largest weight increase (by 104.6%) of bulbs in vitro was noticed in cultures that were grown for 6 weeks at 4 °C. Bulb weight increase is very important for acquiring the competence to undergo the vegetative phase change in bulbous plants because the critical size of the bulb was proven to be more important than its physiological state [28]. A strong correlation exists between developmental stage and size of the bulb in different species. Langens-Gerrits et al. [28] confirmed the importance of bulb size, showing that lily bulblets over 800–900 mg fresh weight always formed a stem under inductive conditions. Many factors affect bulb size during in vitro growth: sucrose concentration in medium, composition of nutrition medium (mineral salts and growth regulators) [44], size of the initial explant [28] and lighting environment [45]. 

Pretreatment with low temperature (4 and 15 °C) during several weeks could promote multiplication of bulbs upon their transfer to 24 °C. Petrić et al. [39] detected increased bulb multiplication after pretreatment with low temperatures. The highest number of bulbs (8.7) was noted for bulbs previously cultured at 4 °C over 8 weeks. Prolongation of cold treatment to more than 8 weeks had a negative effect on the average number of bulbs as well as on bulb weight in *F. meleagris*. The effects of different temperature pretreatments and their duration on bulb multiplication, weight and the percentage of sprouting in *F. meleagris* are presented in Table 2.

The content of photosynthetic pigments was determined in *F. meleagris* bulbs after chilling (4 °C for 6 weeks). The amounts of chlorophyll a and b were more than 6-fold lower in bulbs after low-temperature treatment compared to control bulbs (at 24 °C). However, when chilled bulbs were transferred to 24 °C, their chlorophyll content after one week of cultivation increased, exceeding that in control bulbs [46]. Carotenoid content showed the opposite trend. Decline in levels of photosynthetic pigments was very sharp during cold treatment, but their content increased after that period, when plants started to grow and when dormancy was released. Increased carotenoid content could be related to their potential role in defense of photosystem II from free radicals [47] during low-temperature treatment.

Active developmental processes that take place during bulb dormancy require energy, carbohydrate partitioning and water uptake. These reserves can only be supplied by the underground organ source, and their mobilization and transport are affected by temperature conditions. Low temperature induced water transfer from lateral to central scales, and subsequently enhanced its transfer to developing buds, while at higher (than optimal) temperatures, water transfer to the buds seemed to be inhibited [26,34]. Cold treatment resulted in an increased number of mitochondria in bulbs, and a significant negative correlation between mitochondrial activity and sucrose levels in cooled tulip bulbs was established [21]. The authors argued that the cold treatment caused the rise in sucrose levels and its subsequent relocation to the shoot, which appear to be essential for normal flowering and which are suppressed in non-cooled bulbs [21,48]. The accumulation of dry matter and water bonding in the cells of central bud in tulip bulbs might occur more intensively at low temperatures [23].

The effect of low temperature on dormancy release in vitro was investigated in *Lilium* [28,49], *Allium* [50] and tulip [51]. Duration of the required low-temperature period depended on species and varied even among different varieties of the same species. Optimum exposure to cold temperatures resulted in higher sprouting percentage as well as intensified growth of roots and leaves. Prolongation of cooling time (to longer than optimal) had no influence on the mentioned parameters, and very often had adverse effects.

### 2.2. Influence of Sugars on Bulb Dormancy

Increased concentration of sucrose in nutrition medium could also enhance bulb multiplication in vitro [39,52]. The weight increase of *F. meleagris* bulbs was observed when bulbs were grown on medium with 4.5% sucrose [53]. The weight of bulbs cultured at 24 °C and higher sucrose concentration (4.5%) increased by 40.96% compared to bulbs cultured on medium with standard sucrose concentration (3%). An even greater increase in bulb weight in vitro was noticed on medium with 4.5% sucrose after pretreatment with low temperature (4 °C for 4 weeks): after dormancy release at 4 °C, adoption of sucrose from nutrition medium was intense at 24 °C, and bulb size increased. Higher sucrose concentration (6%) in medium had a negative effect on weight increase at all investigated temperatures. In most ornamental geophytes (*Lilium*, *Tulipa*, *Narcisus*, *Gladiolus*, *Crinum*), elevated sucrose levels (6–10%) induced storage organ formation in vitro [7]. Rodrígues-Falcón et al. [54] suggested that in potato, sugars may induce cell division and cause tuber swelling, thus increasing tuber size during cultivation on medium with increased sucrose concentration. In line with this, increased sucrose concentration in nutrition medium proved to be an important factor for formation, growth and regeneration of bulbs of *F. meleagris* in vitro. Increased sucrose concentration (9%) induced bulb swelling of lily’s bulblets in vitro, but lower concentration promoted differentiation of shoots and roots [55]. These findings indicated that sucrose may play a very important role as a signal molecule in bulb development, besides its role as a major nutrition supplement. Accordingly, increased sucrose concentration in nutrition medium proved to be an important factor in formation, growth and regeneration of bulbs of *F. meleagris* in vitro.

The effects of temperature regime and nutrition medium composition on bulb weight, dormancy release and sprouting in *F. meleagris* are summarized in Table 3. 

### 2.3. Influence of Gibberellic Acid (GA_3_) on Bulb Dormancy

GA_3_ plays an important role in regulation of in vitro storage organ formation in many geophytes, including ornamentals [56]. In bulbous plants, de novo biosynthesis of gibberellins is an essential factor for shoot elongation and sprouting [57]. Exogenously applied gibberellins can partially substitute for the required cold treatment [58,59]. 

Sprouting of bulbs can be enhanced by short treatment with gibberellins, with or without exposure to low temperature [60]. Bulbs of *F. meleagris* grown at 24 °C for 4 weeks after cold pretreatment (4 °C) on medium supplemented with GA_3_ had a higher percentage of regeneration compared to bulbs grown without gibberellin [39]. Culturing bulbs at 4 °C at all tested GA_3_ concentrations (1, 2 and 3 mg L^−1^) had a positive effect on their multiplication and weight. 

GA_3_ also positively influenced sprouting of *F. meleagris* bulbs. Bulbs sprouted when they were treated with gibberellin even if they had not been exposed to low-temperature treatment. However, sprouted bulbs did not continue to grow further unless previously cultured at 4 °C for several weeks. Without exogenously applied gibberellin, bulb formation was stimulated, and dormancy initiated. Contrary to these findings, Kim [61] showed that GA_3_ (at 1 mg L^−1^) had no effect on dormancy initiation in *Lilium speciousum*, while GA_4+7_ (at 1 mg L^−1^) partially prevented the development of dormancy. Such differences could be explained by different gibberellin concentrations used or species-specific physiology.

The growth retardants (retardants of GA_3_ synthesis, ancymidol and paclobutrazol) are very often used in vitro because they stimulate bulb multiplication, reduce leaf and root growth and also prevent hyperhydricity and leaf damage [62]. Of all used growth retardants, paclobutrazol seems to be the most effective. In the presence of this GA inhibitor, leaf and root formation was very much reduced or completely stopped. This phenomenon led to enhanced growth of bulbs in vitro and an increase in their weight. Application of GA_3_ (liquid solution, 1, 10 and 100 mg L^−1^) strongly stimulated shoot multiplication and growth of tulips in vitro [18], similar to the influence on *F. meleagris* bulbs, where tested concentrations were much lower (1, 2 and 3 mg L^−1^) [39]. 

The effect of gibberellin on bulb dormancy and morphogenesis in *F. meleagris* could be monitored by using GA biosynthesis inhibitors [40]. Percentage of sprouted bulbs was higher when bulbs were cultured on plant growth regulator (PGR)-free MS medium, compared to bulbs cultured on medium with GA inhibitors (ancymidol and paclobutrazol). Although inhibitors of GA synthesis should theoretically block sprouting, some of the bulbs sprouted in their presence, mostly because of endogenous GA_3_ content, which was sufficient to initiate sprouting. 

GA_3_ strongly increased the number of sprouted bulbs and their weight, regardless of low-temperature treatment (Figure 2, unpublished results). Bulbs sprouted after GA_3_ application (Figure 2D–F) exhibited the same morphology as control bulbs at either tested temperature, 7 (Figure 2A–C) or 24 °C (Figure 2G). Bulbs cultured with GA_3_ exhibited no signs of necrosis after several weeks at 24 °C (Figure 2H). Marković et al. [40] also showed a significant fresh weight increase when bulbs were grown on medium with inhibitors compared to control, which indicated that GA_3_ did not play a key role in bulb biomass production.

The onset of sprouting, facilitated by GA_3_ application, was correlated with increased sugar accumulation in bulbs [40,63,64]. Soluble sugar accumulation marked the onset of sprouting in *F. meleagris* bulbs [40,64]. Considering that sugar content was lower when GA inhibitors were applied, GA_3_ could be the main responsible factor for sugar accumulation in bulbs. Ranwala and Miller [58] confirmed that de novo biosynthesis of gibberellins in properly cooled tulip bulbs was necessary for proper flower stalk elongation, and that during the rapid elongation phase, gibberellins played an important role in expressing higher acid invertase activity, enabling the cleavage of imported sucrose to hexoses that are utilized in elongating cells. Sugar accumulation in *F. meleagris* was higher when bulbs were cultured at 7 °C without GA_3_ or inhibitors. This indicated that soluble carbohydrates, predominantly sucrose, were inducers of bulb formation [65] at low-temperature conditions, while GA_3_ triggered dormancy release and served as a signal for the onset of sprouting. GA_3_ in culture medium increased sugar reserve in bulbs and enhanced their sprouting, but low temperature was the obligatory factor for further development.

## 3. Sprouting of Bulbs after Dormancy Release

Dormancy prevents sprouting of bulbs in a way that is not fully elucidated. Under in vitro conditions, many bulbs of *F. meleagris* did not sprout without cold treatment, which is necessary for dormancy release [39]. Ledesma et al. [66] noticed that garlic bulbs did not sprout after planting, but their sprouting was extremely increased when they were exposed to low temperature. Garlic bulbs formed in vitro were also dormant and they required a period of low temperature for dormancy breaking [67]. After dormancy release, the vegetative adult phase is completed. In certain flower bulbs (narcissus, iris, tulip), the vegetative adult phase is very short and is immediately followed by the reproductive phase (flower formation), with flower emerging directly from the bulb. This is not the case in fritillary, where sprouting of bulbs marked the ending of vegetative phase and formation of a stem [7]. Langens-Gerrits et al. [28] showed that reorganization of apical meristem, marking the transition from juvenile to adult phase, led to the formation of an actively growing stem primordium in lily. Intensive mitotic activity in the apical meristem layers was correlated with the preparation for sprouting after dormancy release.

Culturing bulbs at low temperature is essential for the formation of a stem. Bulbs must sprout quickly and uniformly for the requirements of the horticulture industry. Knowledge about the regulation of stem formation is very important from a horticultural point of view since bulbs that sprout with a stem after planting grow faster than bulbs that sprout with leaves only [28]. In vitro studies enable researchers to understand the trigger for sprouting of small bulbs, regenerated in culture, planted in soil. Therefore, studies about dormancy and its physiology, control and release are important for better understanding of sprouting and its improvement.

Culturing bulbs of *F. meleagris* at lower and moderate temperatures (4 and 15 °C) had a positive effect on their sprouting compared to control bulbs that were grown constantly at 24 °C [39]. The best results for sprouting of fritillary bulbs were obtained for bulbs previously grown at 4 °C over 4 weeks (Figure 1B). Further prolongation of cold pretreatment (above 4 weeks) had a negative effect on sprouting percentage. The bulb transition into sprouting phase depended on many factors, with bulb size being a major factor that influences the phase transition [10]. However, the relation between bulb growth and phase transition was not absolute, since some external conditions (high sucrose concentration, large explant size, prolonged exposure to low temperature) could stimulate phase transition independently of growth [28]. Cultivation of *F. meleagris* bulbs in vitro, at constant temperature of 15 °C, resulted in development of non-dormant bulbs which sprouted normally [39]. On the contrary, no stem formation occurred in lily bulblets cultured for 12 weeks at 15 °C, with only a low percentage of non-dormant bulbs after 16 weeks of regeneration at 15 °C [28]. As it was shown in the experiment, the temperature shift proved to be crucial for sprouting and further development, with a short culture period at lower temperature being essential for the induction of the juvenile to adult phase change.

Besides dormancy release, cold treatment (4 °C) resulted in an increase in rooting and sprouting of in vitro regenerated bulbs of *F. meleagris*. Number of roots after 6 weeks of chilling was two times higher than in bulbs cultured at 24 °C (Figure 1C). Root induction rate of *F. meleagris* bulbs after six weeks of chilling was 60.4%, while at 24 °C (without cold treatment), it was 32.4%. Elongation rates of both shoots and roots were also higher in bulbs grown at low temperature (Figure 1D). In vitro regenerated bulbs with roots could be acclimatized in the greenhouse. Survival rate of these plantlets was higher when bulbs had been chilled for several weeks prior to transfer to 24 °C [68].

Developed plants of *F. meleagris*, grown in vitro for 10 weeks at 24 °C, showed significant differences in morphological parameters compared to plants obtained from bulbs that were previously cultured for 6 weeks at 4 °C [46]. Average height of plants obtained from bulbs cooled for 6 weeks was higher by 42.92% compared to plants grown without cold treatment. Also, the average number of roots and their length were higher than for plants cultured constantly at 24 °C. Cooling of bulbs induced a higher number of roots in many geophytes. In *Dioscorea polystachya*, the percentage of rooting in cooled bulbs was 99%, compared to the much lower rooting percentage (9%) found in non-cooled bulbs [69]. In vitro propagated bulbs had a very high survival rate, higher than shoots, because bulbs have an optimal surface/volume ratio and are covered with a protective outer layer [56].

Sprouted and non-sprouted bulbs of *F. meleagris*, grown in vitro for 5 weeks at either 7 or 24 °C, had numerous amyloplasts with starch granules, but their number decreased when sprouts continued to grow and enlarged (Figure 3A). After dormancy release, localized meristems became visible (Figure 3B–E). Histological analysis did not reveal a remarkable difference between chilled (Figure 3E, F) and non-chilled, sprouted bulbs (Figure 3A–D). Marković et al. [40] suggested that starch has been utilized as an energy source for sprouting and further development of *F. meleagris* bulbs after dormancy release. 

## 4. Sugar Status of Sprouted Bulbs after Dormancy Release

During the period of dormancy, accumulation of soluble sugars (mainly sucrose, glucose and fructose) was detected in *F. meleagris* bulbs [40,68]. These sugars were necessary for subsequent beginning of sprouting after the end of cold treatment. In *F. meleagris* bulbs, there was no significant accumulation of sugars immediately after the cold treatment but an increase in glucose and total sugars was detected several weeks after chilling. Similar findings were noticed in lily bulbs stored at low temperature [70] when starch started to disappear, and sucrose (energy source for further development) started to accumulate. 

Sugar concentration in *F. meleagris* bulbs, cultured at 7 °C, remained almost unchanged over the course of time, but after chilling, sugar concentration, particularly glucose, increased rapidly [40,64]. Increased glucose accumulation in *F. meleagris* bulbs was detected in the lower sprout portion after the end of the chilling period. This could be a result of starch degradation in bulbs and mark the beginning of sprouting. Glucose concentration did not significantly differ in lower and upper sprout portion, when bulbs were not chilled for a particular time. However, the dramatic increase in glucose concentration after chilling and its predominant accumulation in the bottom part of the leaves during early sprouting indicated a huge influence of chilling on sugar status after dormancy release in *F. meleagris* [64]. Glucose was proven to be a signal molecule that can influence the storage organ formation [71], as shown in peony propagated in vitro, with increased glucose concentration (6–9%) in medium [72]. 

Low temperature leads to intensive starch hydrolysis and sucrose synthetic pathway in many geophytes [10,34]. Sucrose may have a role in phase change of bulblets and the beginning of sprouting by stimulation of cell divisions in apical meristem, as is the case in *Arabidopsis* [73]. The synthesis of carbohydrates is associated with the late phases of bulb formation which occurred during cold treatment in tulip [74]. 

Cold treatment leads to an increased expression of acid invertase genes and high activities of the respective enzymes, which boosts hexose production in growing sprouts [58,75]. Thereby, accumulated carbohydrates will participate in lily shoot elongation in the next vegetative period [75,76]. Bulbs deprived of cold treatment have lower sugar content, as well as the emerging shoots, which therefore display only slight growth after planting [40,74]. 

## 5. Enzymatic Profile and Antioxidative Defense System in Relation to Dormancy

The impact of stress in plant environment can be measured indirectly by measuring activities of antioxidative enzymes, the most common markers of oxidative stress. Those enzymes are a part of the antioxidative plant defense system. Enzymes such as catalase (CAT), peroxidase (POX), superoxide dismutase (SOD) and glutathione reductase (GR) are able to eliminate reactive oxygen species (ROS) accumulated in plant tissues during different kinds of stresses. Low temperature, which is necessary for breaking dormancy, represents some kind of stress for a plant and the activity of those enzymes could be an indicator for different phases of bulb dormancy [75]. 

Increased activity of SOD and POX was detected after 3 days of cold treatment in bulbs of *F. meleagris* in vitro [39]. On the contrary, the activity of GR was low throughout the entire cold treatment period. A similar trend in activity of antioxidative enzymes was noticed in bulbs grown ex vitro. Antioxidative enzymes were shown to have a crucial role in ROS elimination during growth at low temperatures [77]. The activity of all antioxidant enzymes of *F. meleagris*, except GR, increased with decreasing temperature. SOD showed the highest activity after 8 weeks at 4 °C. Certain isoforms of SOD could be associated with dormancy and a period of low temperatures, as it has been documented that genes for specific isoforms were activated at low temperatures [78,79,80]. Manganese superoxide dismutase (Mn-SOD) was detected in bulbs of *F. meleagris* during low-temperature treatment [81]. Vyas and Kumar [82] suggested that this isoform of SOD was involved in elimination of ROS accumulated under stress conditions caused by low temperatures. *F. meleagris* grown in natural conditions over five autumn–winter months exhibited the highest SOD activity, which may be related to the activity of Mn-SOD [39]. 

Similar to SOD, CAT played an important role in oxidative stress induced by low temperatures. CAT activity rapidly increased at the beginning of cold period in bulbs of *F. meleagris*. Many plants which are well-adapted to low temperatures had increased CAT activity, especially at the beginning of cold treatment or the winter season [83,84]. CATs potentially eliminate free radicals accumulated in response to low temperatures, under natural or experimental conditions. One of the major ROS species in bulbs, especially at the beginning of cold treatment (winter period), was hydrogen peroxide, which was accumulated by the activity of SOD. Activity of CAT depended on hydrogen peroxide concentration. Consequently, CAT activity was the highest at the beginning of the cold period in *F. meleagris* bulbs grown at 4 °C [39].

GR activity followed the opposite trend of that observed for SOD and CAT. In *F. meleagris*, its activity was the lowest when bulbs were grown in natural conditions over the 5 months of autumn–winter and the highest after dormancy release. Metabolic processes, strongly activated after winter (after dormancy release), led to accumulation of large amounts of ROS. In *F. meleagris* bulbs, these large amounts of ROS were probably eliminated by GR activation.

POX activity was the highest at the beginning of cold treatment in vitro or the winter period in natural conditions. Peroxidases had the lowest activity when CAT activity was the highest (these enzymes had the opposite action) because they had different and specific affinity for hydrogen peroxide, the concentration of which varied during low-temperature treatment or winter. Both enzymes were active in bulbs of *F. meleagris* at different time points of dormancy, but CAT activity was generally higher. This led us to assume that CAT has a major role in dormancy release and the onset of sprouting in the spring [46]. Trends in enzyme activity of *F. meleagris* under low-temperature treatment are shown in Table 4.

Potential markers for morphogenesis are esterases, characterized as acidic esterases [84]. These enzymes are involved in many physiological processes in plants. Esterases were found in bulbs of *F. meleagris* during cold treatment [53]. In bulbs cultured at 4 °C, esterase isoform with a pI value between 5.20 and 5.85 was detected throughout the entire culture period. Esterase activity was increased during the initial stages of morphogenesis after low-temperature treatment, indicating that esterases may play a role in dormancy release.

Enzyme activity was investigated in bulb scale segments of *F. meleagris* cultured in vitro, during induction of morphogenesis on different nutrition media [46]. Bulbs were previously cultured at 4 °C for 6 weeks. After dormancy release and the beginning of morphogenesis, SOD and CAT activity were the highest. Increased activity of these enzymes during the first weeks of morphogenesis induction following cold treatment could be related to growth regulators in nutrition media and to intensive metabolic processes after dormancy release, caused by low temperature. POX activity showed the opposite trend to those of SOD and CAT. Activity of these enzymes strongly correlated with accumulation of hydrogen peroxide under stress conditions [46]. Hydrogen peroxide may act as a signaling molecule in regulation of a variety of physiological processes [85]. Antioxidative enzymes are known to play a key role in maintenance and release of dormancy in plants [46,81,86]. Therefore, understanding the biochemistry of dormant and sprouting bulbs could facilitate artificial induction and release of bulb dormancy.

## 6. Conclusions

Regeneration of bulbs is a crucial stage in micropropagation of *F. meleagris* and other ornamental geophytes. Dormancy exhibited during the annual life cycle of geophytes is an indispensable condition for their normal development. Physiological aspects of dormancy are still insufficiently explored, but careful examination of dormancy patterns in different species can bring us closer to understanding the underlying mechanisms that control bulb dormancy. Studies on regulation of the growth cycle of bulbous plants concerned processes of storage organ formation, growth and sprouting, all of which are important from horticultural and economic points of view. The whole developmental process of *F. meleagris* takes two growing seasons in natural conditions and it could be modeled in tissue culture. The manipulation with temperature, sucrose concentration and plant growth regulators in medium may accelerate and synchronize bulb formation and dormancy release in vitro.

## Figures and Tables

**Figure 1 plants-10-00902-f001:**
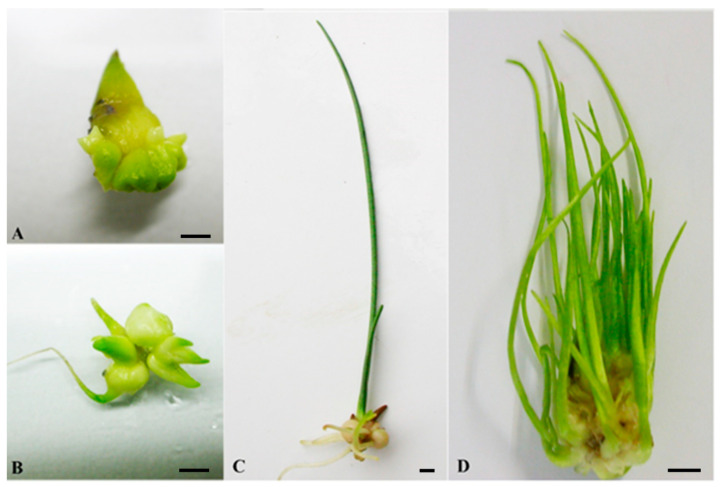
Different stages of in vitro bulb development in *F. meleagris* depending on temperature regime. (**A**) Dormant bulb regenerated on MS medium (4 weeks at 24 °C). (**B**) Sprouted bulb after cold pretreatment (4 weeks at 4 °C). (**C**) Rooted bulb after dormancy release, cultured at 24 °C after cold pretreatment. (**D**) Shoot multiplication, at 24 °C after cold pretreatment. Scale bars = 3 mm.

**Figure 2 plants-10-00902-f002:**
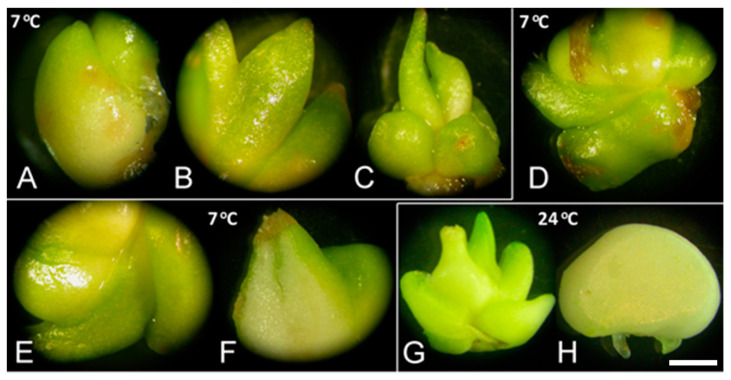
Sprouting of *F. meleagris* bulbs cultured at 7 and 24 °C. (**A**–**C**) Sprouted bulbs grown at 7 °C on culture medium without PGRs after one (**A**), three (**B**) and five weeks (**C**). (**D**–**F**) Sprouted bulbs grown at 7 °C on medium supplemented with 10 μM GA_3_ after one (**D**), two (**E**) and five weeks (**F**). (**G**) Bulbs grown for five weeks at 24 °C on culture medium without PGRs. (**H**) Bulbs grown for two weeks at 24 °C on culture medium supplemented with 10 μM GA_3_. Scale bar = 3 mm.

**Figure 3 plants-10-00902-f003:**
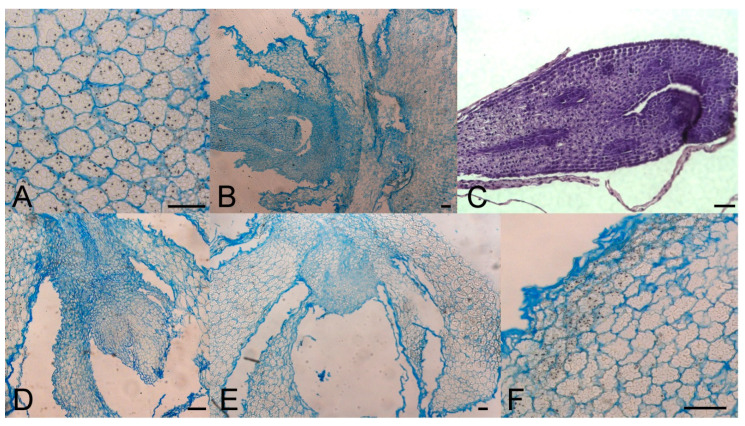
Morpho-anatomical study of *F. meleagris* bulbs sprouting after 5 weeks of culture with GA_3_ and GA inhibitors (10 μM). (**A**) Control bulbs cultured at 24 °C. (**B**–**D**) Bulbs cultured at 24 °C on medium with ancymidol (**B**), paclobutrazol (**C**) and GA_3_ (**D**). (**E**,**F**) Bulbs cultured at 7 °C on medium with GA_3_. Scale bars = 100 μm.

**Table 1 plants-10-00902-t001:** Effect of temperature regime and nutrition media composition on bulb dormancy release in vitro in different *Fritillaria* species.

*Fritillaria* Species	Temperature Regime	Treatment *	References
*F. thunbergii*	5 °C, 5 weeks	MS	[6]
	4 or 10 °C, 75 days	MS	[37]
*F. meleagris*	7 °C, 51–56 days	B5/2 + 5 μM NAA + 0.5 mg L^−1^ activated charcoal	[38]
	4 or 15 °C, 4–10 weeks	MS	[39]
	7 °C, 2–5 weeks or 24 °C, 5 weeks	MS + 10 μM GA_3_	[40]
*F. persica*	4 °C, 75 days + 10 °C, 15 days	MS + 1 mg L^−1^ BAP + 0.1 mg L^−1^ IBA	[41]
	4 °C, 30 days	MS + 50 mg L^−1^ sucrose	[41]
*F. tubiformis*	4 °C, 30–60 days	MS	[42]

* MS—Murashige and Skoog nutrition medium; NAA—naphthaleneacetic acid; BAP—benzylaminopurine; IBA—indole-3-butyric acid; GA_3_—gibberellic acid.

**Table 2 plants-10-00902-t002:** Effect of pretreatment and its duration on *F. meleagris* bulb multiplication, weight and sprouting after 4 weeks of subsequent growth at 24 °C [7,39].

Pretreatment (Duration)	Bulb Multiplication	Bulb Weight	Sprouting of Bulbs
4 °C (4 weeks)	+	+ *	+ (~80%) ^§^
4 °C (6 weeks)	+	+ *	+ (~75%)
4 °C (8 weeks)	+ *	+ *	+ (~70%)
4 °C (10 weeks)	+	+	+ (~60%)
15 °C (4 weeks)	+	+	+ (~85%)
15 °C (6 weeks)	+	+	+ (~85%)
15 °C (8 weeks)	+ *	+	+ (~70%)
15 °C (10 weeks)	+	+ *	+ (~70%)
24 °C (4 weeks)	−	−	± (~30%)
24 °C (6 weeks)	−	−	± (~30%)
24 °C (8 weeks)	−	−	± (~30%)
24 °C (10 weeks)	−	−	± (~30%)

+ positive effect, − no effect, ± bulbs started to sprout but did not continue to grow further. * The best result obtained for defined temperature. ^§^ Approximate percentage of sprouted bulbs after 4 weeks of growing at 24 °C after pretreatment.

**Table 3 plants-10-00902-t003:** Effect of pretreatment and nutrition medium composition on bulb weight, dormancy release and sprouting of *F. meleagris* bulbs [7,37].

Pretreatment (4 Weeks)	Treatment *	Bulb Characteristics
24 °C	3% sucrose	Weight unchanged Dormancy continues No sprouting
4 °C	3% sucrose	Weight unchanged Dormancy terminated Bulbs sprouted
24 °C	4.5% sucrose	Weight increased Dormancy continued No sprouting
4 °C	4.5% sucrose	Weight increased Dormancy terminated Bulbs sprouted
24 °C	1 mg L^−1^ GA_3_	Dormancy terminated Onset of sprouting
4 °C	1 mg L^−1^ GA_3_	Weight increase Higher multiplication rate Dormancy terminated Bulbs sprouted

* Basal MS medium was used in all treatments.

**Table 4 plants-10-00902-t004:** Effect of temperature regime on activity of antioxidant enzymes assayed in *F. meleagris* bulbs in vitro [46].

Temperature Regime	Changes in the Activity of Antioxidant Enzymes			
	SOD	POX	CAT	GR
24 °C (8 weeks)	↘	↘	↗	↗
15 °C (8 weeks)	↘	↗	↘	↗
4 °C (8 weeks)	↗	↘	↗	↘
4 °C (3 days)	↗	↗	↗	↘
4 °C (8 weeks) followed by 24 °C (7 days)	↗	↘	↗	↘

Up-right and down-right arrows denote increase or decrease in enzyme activity, respectively.

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
