# Peer review of "Bulb Dormancy In Vitro—Fritillaria meleagris: Initiation, Release and Physiological Parameters"

_plants, 2021, doi:10.3390/plants10050902_

Round 1

Reviewer 1 Report

There are some mistakes in numbering the references e.g.

Reference: Bach, A.; Kapczyńska, A.; Dziurka, K.; Dziurka, M. Phenolic compounds and carbohydrates in relation to bulb formation in 575
Lachenalia ‘Ronina’ and ‘Rupert’ in vitro cultures under different lighting environments. Sci. Hortic. 2015, 188, 23-29 - should have number 43 in the refeference list. I suggest to check all the references.

Author Response

Response to Reviewer 1 Comments

There are some mistakes in numbering the references e.g.

Reference: Bach, A.; Kapczyńska, A.; Dziurka, K.; Dziurka, M. Phenolic compounds and carbohydrates in relation to bulb formation in 575
Lachenalia ‘Ronina’ and ‘Rupert’ in vitro cultures under different lighting environments. Sci. Hortic. 2015, 188, 23-29 - should have number 43 in the refeference list. I suggest to check all the references.

Thank you very much for reading the manuscript plants-1189729. Your suggestion regarding reference numbers was corrected in the revised version of the manuscript.

Reviewer 2 Report

The work was improved,
it is necessary that the authors improve the quality of the figures, include a measurement parameter to compare with others.

There are references that are still outdated, the authors can include other current ones.

I consider that now can be accepted If the authors take these suggestions into account.

Author Response

Response to Reviewer 2 Comments

The work was improved,
it is necessary that the authors improve the quality of the figures, include a measurement parameter to compare with others.

There are references that are still outdated, the authors can include other current ones.

I consider that now can be accepted If the authors take these suggestions into account.

Thank you very much for reading the manuscript plants-1189729. We tried to improve the quality of the figures and we added scale bars on all figures.

Reviewer 3 Report

Fritillaria meleagris is an important flowering plant of the Eurasian species of Liliaceae. The authors have well-tried to compile useful information regarding the Bulb dormancy, its initiation, release and also the physiological parameters associated. The authors must change time to a more understandable one, and a careful English Language check is also required.

Author Response

Response to Reviewer 3 Comments

Fritillaria meleagris is an important flowering plant of the Eurasian species of Liliaceae. The authors have well-tried to compile useful information regarding the Bulb dormancy, its initiation, release and also the physiological parameters associated. The authors must change time to a more understandable one, and a careful English Language check is also required.

Thank you very much for reading the manuscript plants-1189729. We tried to improve the quality of the text and carefully checked the language.

This manuscript is a resubmission of an earlier submission. The following is a list of the peer review reports and author responses from that submission.

Round 1

Reviewer 1 Report

This paper contains an eclectic mixture of previous reports on bulb dormancy (mainly of the last century) and data of the authors.

The same research group published two papers on the same subject in Plants in 2020, and they are now trying to generalize the topic.

I reviewed only 6 pages out of 15, and recommend rejecting this article for several reasons:

  1. The authors call this article a “review”, but in fact, it also contains original data and some repeat publications from 2020, with little variations. If the data is original, then materials and methods should be provided and so this paper is not a review. If the data has already been published, references should be provided. This “hybrid” approach is not clear.
  2. Most references are very outdated. The authors are not familiar with recent publications on geophyte dormancy and geophyte biology, at least as is apparent in this paper.
  3. The tables contain rather “vague” data with no references
  4. The figures are of poor quality, with no scale. I especially objected to Fig. 3. It is a poor example of anatomic work and interpretation.

Some additional comments are also presented in the first pages of the attached manuscript.

If the authors wish to re-write this paper and then to re-submit it in the future, they will have to update references and analyze them, and not just to repeat previous  data.

I recommend full rejection of the manuscript in this form.

Author Response

Responce to Reviewer 1

Thank you very much for reading the manuscript plants-1114791. 

Authors

Reviewer 2 Report

See  attached file.

Author Response

Response to Reviewer 2 Comments

Thank you very much for reading the manuscript plants-1114791. Your comments and suggestions are very much appreciated, as they helped us improve the text. All suggestions made by the reviewer were accepted, and modifications are clearly visible in the track changes version of the revised manuscript plants-1114791. Comments concerning these modifications are given below.

Line 23: Instead dormancy write: dormant period; instead Fritillaria meleagris write: fritillary, Liliaceae

 Keywords have been changed according to the reviewer’s suggestion (‘dormant period’ instead of dormancy; ‘fritillary’ and ‘Liliaceae’ instead of Fritillaria meleagris)

Line 29: geophytes - not only bulbs, also corms, tubers, rhizomes

We have used the term bulb here in its broader sense, according to the statement of Khodorova, N.V., Boitel-Conti, M. [The Role of Temperature in the Growth and Flowering of Geophytes. Plants 2013, 2, 699-711]: “Although the underground organs are classified as true bulbs, corms or tubers of different types (stem, root or hypocotyl), according to the species, a broader definition for the term “bulb” can be applied to all geophytes, whether they are bulbous, tuberous or herbaceous [1].”, who cited the work of De Hertogh and le Nard (1993): [1] De Hertogh, A.A.; le Nard, M. The Physiology of Flower Bulbs: A Comprehensive Treatise on the Physiology and Utilization of Ornamental Flowering Bulbous and Tuberous Plants; Elsevier Science Publishers: Amsterdam, The Netherland, 1993; p. 812]. 

Line 35: the names of botanical families are no longer italicized

Italics has been removed from the name of the family.

Line 83: dormancy of seeds and bulbs or only of bulbs

Table 1 outlines different protocols used for dormancy release of bulbs in different Fritillaria species. Although some papers referred to in the table (e.g. Ref [24], F. persica) deal with seed dormancy as well, in this table only dormancy release concerning bulbs was presented. We have further underlined this by adding ‘bulb’ to the Table caption.

Line 96: how seed germination can be affected by bulb size?

We have mistakenly used the word ‘germination’ instead of ‘sprouting’, which has now been corrected in the main text.

Lines 106-107: what do you mean by optimal conditions (if the bulbs do not grow in such conditions it means that these conditions are not optimal)? for me, the optimal conditions are those that are best for the plant in a given growth phase. Perhaps it is better to write that under favorable conditions, which most often favor the overall growth of plants (but as it turns out, they do not work during the plant dormancy)

The term ‘optimal conditions’ has been changed into “favorable conditions for the overall plant growth”

Lines 156-157 How could you explain it?

[Kim [44] showed that GA3 (at 1 mg L-1) had no effect on dormancy initiation in Lilium speciousum, while GA4+7 (at 1 mg L-1) partially prevented the development of dormancy. ]

Kim et al. used different gibberellin at different concentrations. Also, their experiments were conducted with different geophyte species. Our assumption was that such a differences could be explained by different gibberellin concentrations used or species specific physiology. That was the best we could conclude without further experiments.

Lines 175-179 Your article is a Review, but here (if I understood it correctly) you give your own results, so please mark e.g. in brackets (Authors own results not published), or inform the reader about it in some other way

We added “unpublished results” in the text as well as below the figures.

Lines 183-186 your own results?

Lines 232-235 Is thes your results or somene else, if someone else give reference

Lines 343-345 Again, give the reference or write that this is your results

We added “unpublished results” in the legend to Figures 1 (Lines 183-186), 2 (Lines 232-235) and 3 (Lines 343-345).

Lines 204-205 Not all, see african bulbous plants

We narrowed this statement down to bulbous plants of temperate region, which is now clearly indicated in the text.

Line220 Is 15C low temperature? rather moderate temperature

“Low temperature (4 and 15 °C)” has been changed into “Low and moderate temperature (4 and 15 °C)”

We used the term “low temperatures” for both temperatures (4 and 15) in our previous studies. The term  ‘normal’ or ‘standard’ temperature was used only for the temperature of 24 °C. Here we changed the text according to your suggestion and added moderate temperature, in order to ensure better understanding.

Line 230 what about light condition which are very important in in vitro cultures, so add: and lighting environments; I have found such reference: Bach A., Kapczyńska A., Dziurka K., Dziurka M. 2015. Phenolic compounds and carbohydrates in relation to bulb formation in Lachenalia ‘Ronina’ and ‘Rupert’ in vitro cultures under different lighting environments. Scientia Horticulturae 188: 23–29.

We added the suggested factor influencing bulb size, as well as the corresponding reference (53): “Many factors affect bulb size during in vitro growth: sucrose concentration in medium, composition of nutrition medium (mineral salts and growth regulators) [52], size of the initial explant [31], as well as lighting environment [53]. “

Lines 241-243 Again, this is your own results, so inform about it.

Results presented in Table 3 are our own results, from our previous studies. We added references to these previously published papers to the Table 3 caption.

Line 279 as I noticed you do not focuse on seeds cultivation in invitro, so maybe it would be bettrer not mention about the subject of seeds as you do not refer to this issue further in the text .

We have removed the entire phrase concerning seed germination.

Line 301 moderate temperature

“Culturing bulbs of F. meleagris at lower temperatures (4 and 15 °C)” has been changed into “Culturing bulbs of F. meleagris at lower and moderate temperatures (4 and 15 °C)”

Lines 373-374 give Reference

References were added.

Lines 376-383 give Reference

Reference added.

Lines 408-422 Your results?

These are the results of our previous study. The corresponding reference (54) was added in the text as well as in the Table 4 caption.

Lines 442-443 Reference

We added references in the text.

Reviewer 3 Report

The manuscript entitled “Bulb dormancy in vitro - Fritillaria meleagris: initiation, release and physiological parameters” contains interesting information of both fundamental and practical interest. Therefore I recommend it to be published in the Plants Journal. In order to do so, however serious major issues need to be addressed.

A general remark to the presentation and structuring of the whole manuscript: please re-work in order to distinguish clearly what has been done in literature on other species of bulbous plants in conventional conditions, what has been done of Fritillaria in conventional conditions. Then what has been done in literature on bulbous plants in vitro, what has been done of Fritillaria in vitro. Please, pay special attention references which support work done on other bulbous species not to be understood as information on Fritillaria species.

Please, describe abbreviations throughout the text upon first mention. In the specific case of Table 1, abbreviations can be explained in the legend.

Line 102: Please, write which regulators were applied in the mentioned ranges.

Line 103: Please, give the concrete temperature ranges.

The sub-units of 2. “Dormancy” need to be re-structured and information within better organized. As it is now sub-unit 2.1. is entitled “Dormancy development and factors influencing dormancy release in vitro” and discusses concretely cold treatment and sugars roles in breaking dormancy. On the other hand the next 2.2. is concretely entitled “Influence of GA3 on bulb dormancy” and discusses concretely the role of gibberellins in breaking dormancy. Please separate 2.1. into the respective partitions for each factor, to be similar as 2.2. Table 2, which is placed into the gibberellins’ sub-section 2.2. discusses information also on temperature and sucrose. Please, re-organize the information so that the respective information fits into the respective structuring of the text. In addition, Table 2. lacks the respective references, so it is difficult to correlate with the text. Further on, the 2.3. sub-unit also discusses on the effect of temperature on dormancy. Please try to bring more order into the structuring of the text.

Lines 159-161 - growth retardants discussed here need to be concretely named by the examples of which retardants affected which species, given in the cited work, since the manuscript is a review paper, aiming at providing concrete information. The cited Reference 45.: “Ziv, M. Simple bioreactor for mass propagation of plants. Plant Cell Tiss. Org. 2005, 81, 277-285” mentions very shortly the roles of growth retardants (amongst many other factors) on biomass formation in bioreactors in many species. It is not clear, for example, where all the information in paragraph Lines 159-161 comes from (for example the described effects of GA3 on F. meleagris or tulip are not in the cited reference 45).

Although reference 28 has been cited, it has to be better explained in the text that Figure 1 illustrates the authors own work, discussed in the current review. Give the source citation in the Figure legend or if in the text - just next to the citation of Figure 1.

As already discussed above, sub-unit 2.2. deals with the effect of temperature on dormancy, which is also present in 2.1.

Similarly as above, please indicate that Figure 2 presents results of the authors own current work. Give the source citation in the Figure legend or if in the text - just next to the citation of the Figure 2.

Lines 214-215: “Fritillary bulbs regenerated in vitro could not continue to grow and develop normally without having been subjected to low temperature, similar to many other geophytes [51].” Please check the cited reference. In it Fritillaria has not been discussed.

Line 280: Please give citation for the statement: “Under in vitro conditions, many bulbs of F. meleagris did not sprout without cold treatment, which is necessary for dormancy release.”

Line 288: “This is not the case in fritillary where sprouting of bulbs marks the ending of vegetative phase and formation of a stem.” Please, support with a citation. Please mention which species is discussed in Reference 31 following after the latter statement.

Sub-unit 4. Sugar status of sprouted bulbs after dormancy release

The whole description of sugars pools in bulbs of Fritillaria and other species (in vivo, in vitro) is not very understandably presented.

Just an example: the following paragraphs need more clarity in order to understand the character and roles of sugars being accumulated during dormancy (according to the whole previous content of the manuscript dormancy and cold treatment are closely related) “During the period of dormancy, accumulation of soluble sugars (mainly sucrose, glucose and fructose) was detected in F. meleagris bulbs [61,28]. These sugars were necessary for subsequent beginning of sprouting after the end of cold treatment. When lily bulbs were stored at low temperature, starch started to disappear and sucrose started to accumulate [63].In F. meleagris bulbs, there was no significant sucrose accumulation after cold treatment, but both glucose and fructose accumulation was detected and their concentration was lower in chilled bulbs, which supports the finding that sucrose is the main sugar in dormant bulbs [47].”

In this section it cannot be clearly understood also which of the published information concerns sugars accumulation after dormancy in Fritillaria in vitro.

Sub-unit 5. Enzymatic profile and antioxidative defense system in relation to dormancy

Lines 376-382: Please, support with references.

Author Response

Response to Reviewer 3 Comments

The manuscript entitled “Bulb dormancy in vitro - Fritillaria meleagris: initiation, release and physiological parameters” contains interesting information of both fundamental and practical interest. Therefore I recommend it to be published in the Plants Journal. In order to do so, however serious major issues need to be addressed.

A general remark to the presentation and structuring of the whole manuscript: please re-work in order to distinguish clearly what has been done in literature on other species of bulbous plants in conventional conditions, what has been done of Fritillaria in conventional conditions. Then what has been done in literature on bulbous plants in vitro, what has been done of Fritillaria in vitro. Please, pay special attention references which support work done on other bulbous species not to be understood as information on Fritillaria species.

Please, describe abbreviations throughout the text upon first mention. In the specific case of Table 1, abbreviations can be explained in the legend.

The abbreviations are now all described upon their first mention in the text, including those used in Table 1, which have been explained in the footnote.

Line 102: Please, write which regulators were applied in the mentioned ranges.

We listed all the applied plant growth regulators in the text.

Line 103: Please, give the concrete temperature ranges.

We added concrete temperature ranges in the text.

The sub-units of 2. “Dormancy” need to be re-structured and information within better organized. As it is now sub-unit 2.1. is entitled “Dormancy development and factors influencing dormancy release in vitro” and discusses concretely cold treatment and sugars roles in breaking dormancy. On the other hand the next 2.2. is concretely entitled “Influence of GA3 on bulb dormancy” and discusses concretely the role of gibberellins in breaking dormancy. Please separate 2.1. into the respective partitions for each factor, to be similar as 2.2. Table 2, which is placed into the gibberellins’ sub-section 2.2. discusses information also on temperature and sucrose. Please, re-organize the information so that the respective information fits into the respective structuring of the text. In addition, Table 2. lacks the respective references, so it is difficult to correlate with the text. Further on, the 2.3. sub-unit also discusses on the effect of temperature on dormancy. Please try to bring more order into the structuring of the text.

Sub-unit 2. Dormancy has been re-structured in a more comprehensible manner, so we hope that the information contained within is now better organized.

We have grouped all information concerning the influence of temperature into one subsection (now 2.1. The role of temperature in bulb dormancy initiation and release). This section also contains Figure 1 (previously entitled Figure 2).

We have separated information concerning the influence of sugars into a subsection entitled 2.2 Influence of sugars on bulb dormancy, in which we also included Table 2 - given the new arrangement, it is now entitled Table 3 (whereas Table 3 is now entitled Table 2), and we also added the respective references in the Table caption.

The subsection 2.3. Influence of gibberellic acid (GA3) on bulb dormancy remained unchanged with respect to text structure, except that the original Table 2 was removed from it, as suggested by the reviewer.

Lines 159-161 - growth retardants discussed here need to be concretely named by the examples of which retardants affected which species, given in the cited work, since the manuscript is a review paper, aiming at providing concrete information. The cited Reference 45.: “Ziv, M. Simple bioreactor for mass propagation of plants. Plant Cell Tiss. Org. 2005, 81, 277-285” mentions very shortly the roles of growth retardants (amongst many other factors) on biomass formation in bioreactors in many species. It is not clear, for example, where all the information in paragraph Lines 159-161 comes from (for example the described effects of GA3 on F. meleagris or tulip are not in the cited reference 45).

We explained growth retardants from study (Ziv, 2005). Also, we changed references from 45 to 15 (regarding tulip), which was an obvious mistake in the text. We added our reference 25 about gibberellin concentration, concerning F. meleagris.

Although reference 28 has been cited, it has to be better explained in the text that Figure 1 illustrates the authors own work, discussed in the current review. Give the source citation in the Figure legend or if in the text - just next to the citation of Figure 1.

We added explanation about our results in form of picture (our unpublished results, which is now stated in the Figure 1 legend).

As already discussed above, sub-unit 2.2. deals with the effect of temperature on dormancy, which is also present in 2.1.

We changed the order of subunits and their subtitles. We divided these two subsections in order to divide the very beginning and further growth. Temperature is part of both subsections, but its influence was divided because the text (described influence before and after) would be too long for one subsection only.

Similarly as above, please indicate that Figure 2 presents results of the authors own current work. Give the source citation in the Figure legend or if in the text - just next to the citation of the Figure 2.

We added “unpublished results” in the Figure legend, according to the reviewer’s suggestion.

Lines 214-215: “Fritillary bulbs regenerated in vitro could not continue to grow and develop normally without having been subjected to low temperature, similar to many other geophytes [51].” Please check the cited reference. In it Fritillaria has not been discussed.

Reference [51], concerning Allium species, was used to describe similarity [of fritillary] with geophytes other than fritillary. However, for clarity we have specified this by adding “such as Allium” prior to citing the reference [51], which describes Allium species and their dormancy release depending on bulb mass..

Line 280: Please give citation for the statement: “Under in vitro conditions, many bulbs of F. meleagris did not sprout without cold treatment, which is necessary for dormancy release.”

Reference concerning F. meleagris bulbs was added.

Line 288: “This is not the case in fritillary where sprouting of bulbs marks the ending of vegetative phase and formation of a stem.” Please, support with a citation. Please mention which species is discussed in Reference 31 following after the latter statement.

We added the required reference. We also provided species name in the following statement, regarding Reference 31.

Sub-unit 4. Sugar status of sprouted bulbs after dormancy release

The whole description of sugars pools in bulbs of Fritillaria and other species (in vivo, in vitro) is not very understandably presented.

Just an example: the following paragraphs need more clarity in order to understand the character and roles of sugars being accumulated during dormancy (according to the whole previous content of the manuscript dormancy and cold treatment are closely related) “During the period of dormancy, accumulation of soluble sugars (mainly sucrose, glucose and fructose) was detected in F. meleagris bulbs [61,28]. These sugars were necessary for subsequent beginning of sprouting after the end of cold treatment. When lily bulbs were stored at low temperature, starch started to disappear and sucrose started to accumulate [63].In F. meleagris bulbs, there was no significant sucrose accumulation after cold treatment, but both glucose and fructose accumulation was detected and their concentration was lower in chilled bulbs, which supports the finding that sucrose is the main sugar in dormant bulbs [47].”

In this section it cannot be clearly understood also which of the published information concerns sugars accumulation after dormancy in Fritillaria in vitro.

We modified the text and we hope it is clear now. We separated the parts concerning F. meleagris and those concerning other plant species described in the section. Also, we added some plant species specially connected with particular descriptions.

Sub-unit 5. Enzymatic profile and antioxidative defense system in relation to dormancy

Lines 376-382: Please, support with references.

Reference that supports statements given in lines 376-382 was added.

Reviewer 4 Report

This is a review that focuses Fritillaria meleagris L. (Liliaceae),  a valuable, bulbous, ornamental plant. The authors realize an analysis of the effect of plant growth regulators and low-temperature pretreatment on dormancy release in relation to the induction of antioxidative enzymes activity in vitro.

Author Response

Responce to Reviewer 4

Thank you very much for reading the manuscript plants-1114791. 

Authors

Round 2

Reviewer 1 Report

This is my second revision of the presented paper, and I still suggest rejecting it.

I will send the .pdf file to the Plants, since I am not able to attach it to this review

  • This paper contains an eclectic mixture of previous reports on bulb dormancy (mainly of the last century) and data from the previous publications of the same research team (refs 37, 32, 39).
  • The authors call this article a “review” but, in fact, it also contains original data and some repeat publications from 2020, with little variations. If the data is original, then materials and methods should be provided, and thus this paper is not a review. If the data have already been published, references should be provided. This “hybrid” approach is not clear to me.
  • Most references are very outdated. The authors are not familiar with recent publications on geophyte dormancy and geophyte biology, at least as is apparent in this paper. Also, recent reports on potato, onion and tree buds contain a lot of new information and can be relevant in this review.
  • The figures are of poor quality, with no scale. I especially objected to the microscopic images. It is a poor example of anatomic work and interpretation.

Round 3

Reviewer 1 Report

This is my third revision of this submission, and I still think that this paper is not suitable for publication. I revised only a few pages of the manuscript. It was improved, but most references are still outdated, and the authors are not familiar with latest publications and modern geophyte science. 

It is not clear from the text - what is the difference between whole plant (in vivo) and explants in vitro, if any. 

There are many repetitions. 

I still think that the Figures are of bad quality, especially histological images.
